# Lifestyle-Informed Personalized Blood Biomarker Prediction via Novel Representation Learning

A. Ali Heydari[†,*], Naghmeh Rezaei[†], Javier L. Prieto[†], Shwetak N. Patel[†], Ahmed A. Metwally[†]

*Abstract*—**Blood biomarkers are an essential tool for healthcare providers to diagnose, monitor, and treat a wide range of medical conditions. Current reference values and recommended ranges often rely on population-level statistics, which may not adequately account for the influence of inter-individual variability driven by factors such as lifestyle and genetics. In this work, we introduce a novel framework for predicting future blood biomarker values and define personalized references through learned representations from lifestyle data (physical activity and sleep) and blood biomarkers. Our proposed method learns a similarity-based embedding space that captures the complex relationship between biomarkers and lifestyle factors. Using the UK Biobank (257K participants), our results show that our deep-learned embeddings outperform traditional and current state-of-the-art representation learning techniques in predicting clinical diagnosis. Using a subset of UK Biobank of 6440 participants who have follow-up visits, we validate that the inclusion of these embeddings and lifestyle factors directly in blood biomarker models improves the prediction of future lab values from a single lab visit. This personalized modeling approach provides a foundation for developing more accurate risk stratification tools and tailoring preventative care strategies. In clinical settings, this translates to the potential for earlier disease detection, more timely interventions, and ultimately, a shift towards personalized healthcare.**

*Index Terms*—**Biomarkers, Deep Learning, Lifestyle, Personalized Predictions, Representation Learning.**

## I. INTRODUCTION

Blood biomarkers are used in disease diagnosis (e.g. glycated hemoglobin [HbA1c] to diagnose diabetes), or determining specific therapies (e.g. prescribing cholesterol-lowering medication to people with hypercholesterolimia). A critical step in the clinical use of blood biomarkers is establishing reference values and recommended ranges (RR) for each biomarker. Currently, a majority of traditional references do *not* take into account age, sex, or other individual factors, which is considered a meaningful limitation [1], [2]. A growing body of evidence suggests that personalizing references can lead to more accurate and reliable diagnoses and better patient outcomes: For example, Zaninetti *et al.* [1] showed the impact of personalizing RRs on the number of people diagnosed with unexplained thrombocytopenia. Rappoport *et al.* [3] compared the differences in laboratory tests based on ethnicity and found the distributions of more than 50% of blood biomarker RRs to differ among self-identified racial and ethnic groups. Beutler *et al.* [4] presented age- and ethnicity-adjusted RRs for hemoglobin and showed the impact on diagnosing anemia. Cohen *et al.* [2] found significant

differences in numerous blood biomarker distributions for numerous laboratory tests based on age and biological sex. Though such studies highlight the significance of personalized RR based on demographics, none account for the impact of lifestyle (e.g. physical activity and sleep) on biomarker reference values, despite several studies reporting the impact of lifestyle on blood biomarekrs in lowering glucose [5], [6], increasing insulin sensitivity [7], [8], and improving lipid profiles [9], [10].

While the significance of personalizing disease management based on lifestyle factors has been the subject of numerous studies (see review paper by Minich *et al.* [11]), research on personalizing blood test ranges based on individual factors, such as lifestyle, are rare [11]. Similar to our goal of predicting future (next visit) blood biomarker values, Cohen *et al.* [2] investigate the effect of personalizing biomarker reference values and propose three strategies for predicting an individual's biomarker values forward in time using untransformed representations (normalized raw data), which, importantly, do not utilize individuals' lifestyle. These strategies are: (1) "Single-Lab" (marker of interest), Single Time Point: Using previous laboratory visits, the authors take the average age- and sex-adjusted laboratory values to perform a 2-year forward regression. (2) "Multi-Lab" (all available blood biomarkers), Single-Time Point: The authors use all available laboratory values and patients' current age to predict the value for the marker of interest two years forward in time.(3) "Multi-Lab, Multi Time Point": The authors use multiple biomarker time points for the prediction of future values. Given our goal of predicting future blood test values from a single visit, the "Multi-Lab, Multi Time Point" technique of Cohen *et al.* is not relevant to our work.

Capturing the relationship between biomarkers and lifestyle factors is crucial for developing effective computational predictive strategies [12]. Recent evidence suggests that the interplay between various biomarkers and lifestyle factors is complex [13], which highlights the importance of considering nonlinear dependencies between various biomarkers and lifestyle. As a result, the field has shifted towards utilizing machine learning (ML) methods to model health and patient outcomes [2], [14]. Recent studies have shown promising results in employing deep learning (DL) approaches that outperform traditional ML techniques in many EHR-related tasks [15], [16]. Improvements have typically been attributed to better mathematical representation of EHR due to DL models' capacity to learn nonlinear mappings from the original untransformed space to an embedding space. Currently, a majority of

† Consumer Health Research, Google LLC, Mountain View, CA 94043, USA (*Corresponding Author: aliheydari@google.com).

DL models for EHR are developed for time series healthcare data. Though these methods work well for cases with years of follow-ups and complete records, the importance of early interventions and high patient churn rate [17]–[20] highlight the need for models that predict future outcomes from a single-visit that can aid in diagnosis or disease prevention. Additionally, many current DL models do not exploit individuals' conditions to produce discriminative embeddings based on similarities between conditions and comorbidities [15], [21]. Deep metric learning (DML) models, such as the *Siamese* [22] and *Triplet* networks [23], have been used extensively in many fields to learn embeddings based on distance (similarity) comparisons. Despite their success in other domains, DML-based models in health-related applications are nascent and under-explored, and the few existing algorithms often focus on medical images rather than actual EHR data [24].

The lack of EHR-specific DML models constitutes a major limitation as blood tests are very different from other typically-used data for DML (such as images or natural language). To address this issue, we introduce a novel DML-based technique for learning patient embeddings. Learning patient similarities from the population can provide additional interpretations and personal insights regarding an individual's blood test values that otherwise may not be available. We benchmark our proposed approach against state-of-the-art approaches, including the EHR-specific model DeepPatient [25]. Though most DL methods in this area consider temporal components, DeepPatient does not explicitly account for time, making it appropriate for comparison with our proposed approach. DeepPatient is an unsupervised DL model that learns general representations by employing three stacks of denoising autoencoders that learn hierarchical regularities and dependencies through reconstructing a masked input of EHR features.

In summary, the main contributions of our work are as follows:

- We investigate the association between lifestyle factors, specifically activity and sleep, on reference blood biomarker values in healthy individuals, determining whether the associations are significant for our model.
- We propose a novel deep metric learning framework that captures complex relationships between blood biomarkers and lifestyle factors to determine personalized blood biomarker references.
- We demonstrate that adding lifestyle factors improve existing single-time blood biomarker models showing the importance of lifestyle factors in clinical applications.
- We show that using our deep-learned representations achieves the highest accuracy, signifying the importance of high-quality representation of individuals as well as lifestyle for predicting future blood biomarker values from a single time point.

## II. METHODS

As illustrated in Fig. 1(a), our proposed approach for single-time prediction of future blood biomarkers and personalized references consists of two steps: The first step is to learn a metric-based mapping that can accurately embed individuals on a latent space based on the similarity of their blood biomarkers, demographics, and lifestyle factors. The next stage is to use the learned embeddings in combination with the current biomarker of interest to predict the future values, which can be used as a personalized reference in the next visit.

### A. Novel Deep Metric Learning Approach for Learning Patient Similarity

We introduce a valuable modification to the traditional triplet loss with the goal of producing embeddings that are more compact for each class and well-separated from dissimilar classes. Although our formulation is focused on EHR, our modified triplet loss can be applied to other domains as well. **Model Formulation**. We aim to learn a transformation $\phi$ that can map similar data points closer together (*e.g.* subjects with comorbidities), and dissimilar data points farther apart (*e.g.* healthy and unhealthy subjects). To do so, we leverage the traditional triplet framework and propose a novel objective based on distances between triplets of data points (anchor, positive, negative).

Let $X$ be the collection of features for individuals with $c \in \mathbb{N}$ many distinct conditions (e.g. diabetes, cancer, etc.). Let $p_i$ and $a_i$ represent the traditional positive and anchor points that are chosen at random from the same class, while the negative vector $n_i$ is chosen to be from a different class randomly (*i.e.* positive or anchor cannot have the same labels as negative). The reasoning behind such selection is to make the naive assumption that all conditions are completely dissimilar, which in turn would force the model to better learn similarities between conditions and comorbidities on its own. Using the selected vectors, and given a margin $\epsilon_0 > 0$ (a hyperparameter), the traditional objective for a triplet network would be formulated as:

$$\mathcal{L}_{Triplet}(p_i, a_i, n_i) = [d(\phi(a_i), \phi(p_i)) - d(\phi(a_i), \phi(n_i)) + \epsilon_0]^+, \tag{1}$$

with $\phi$ denoting a neural network, $[\cdot]^+ = \max\{\cdot, 0\}$, and $d(\cdot)$ being the Euclidean distance. This objective would force the model to satisfy $d(\phi(a_i), \phi(n_i)) > d(\phi(a_i), \phi(p_i)) + \epsilon_0$.

However, given that healthcare records often suffer from high in-class variability within each condition, enforcing additional restrictions on the distance between positive and negative samples could improve learning. Moreover, given our goal to encourage learning similarities from different conditions, we must also enforce additional regularization between dissimilar samples. However, minimizing Eq. (1) only ensures that the negative pairs fall outside of an $\epsilon_0$-ball around $a_i$, while bringing the positive sample $p_i$ inside of this ball, which may be insufficient. For example, many of the healthy patients (those without a clinical diagnosis) may be close to the *bona fide* healthy group, but others in the healthy population may have undiagnosed conditions, therefore being closer to other participants with clinical diagnosis. As a result of the underlying biology and the shortcomings of traditional triplet objective, we add a regularization term with the goal of addressing these two issues, as shown in Eq. (2):

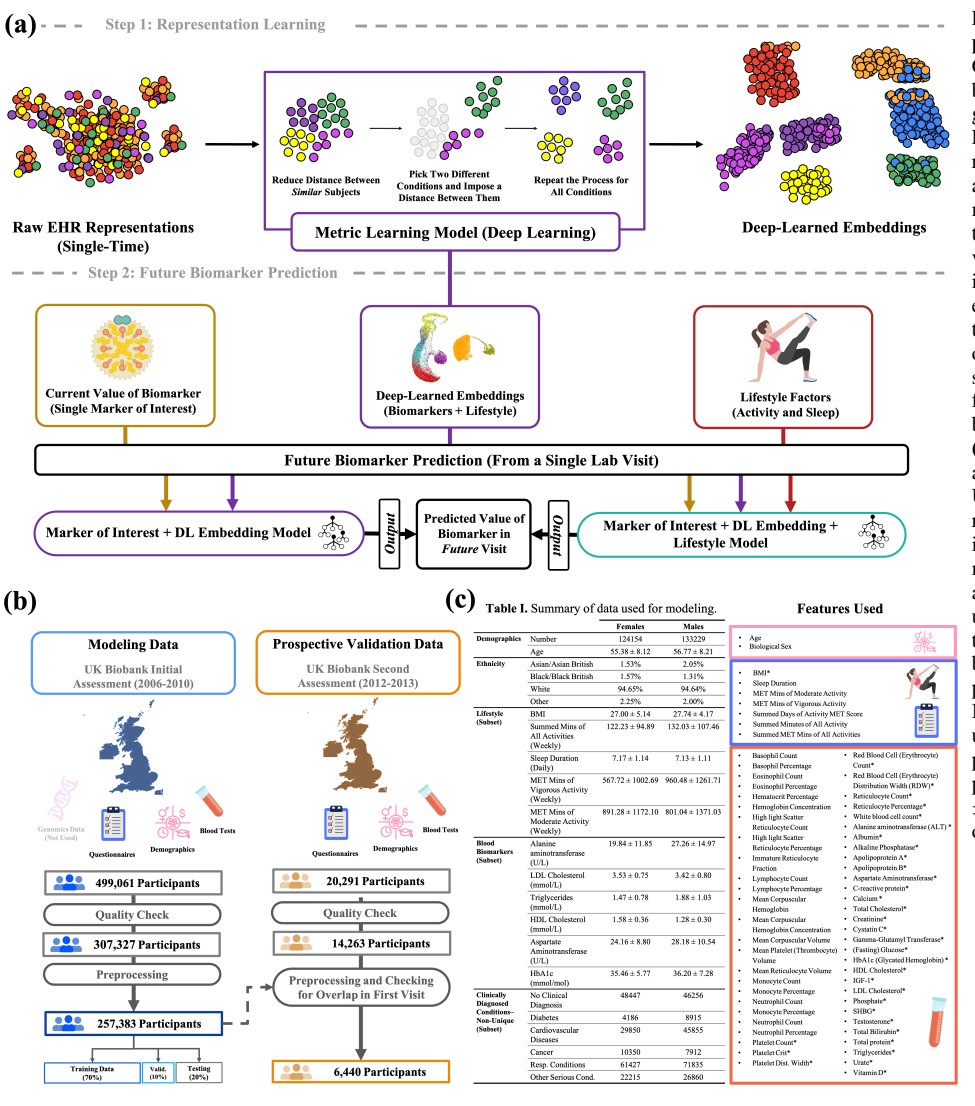

Fig. 1. **Overview of our proposed methodology and data**. **(a)** Our approach for predicting future blood biomarker values from a single lab visit consists of two steps: First, we learn a similarity-based representation of blood biomarkers and lifestyle factors using our novel metric learning technique in order to uncover associations between various health factors. Second, using the learned similarity-based embeddings in combination with the current value of the biomarker of interest, we train biomarker-specific models for predicting the future biomarker values, which can be used as a *personal reference*. **(b)** To showcase our approach on a broad population, we use the United Kingdom Biobank [26]. For representation learning and modeling, we leverage the first assessment (visit), and for assessing the accuracy of future predictions, we utilize the next visit as the prospective validation of our personalized blood biomarker models. **(c)** We present the data summary in Table I, and provide the complete list of used features. Data statistics are presented as number of instances or percentages for counts, or as mean $\pm$ standard deviation for continuous values.

$$\mathcal{L}(p_i, a_i, n_i) = [d(\phi(a_i), \phi(p_i)) - d(\phi(a_i), \phi(n_i)) + \epsilon_0]^+ + [d(\phi(p_i), \phi(n_i)) - d(\phi(a_i), \phi(n_i))]^p \quad (2)$$

where $p \in \mathbb{N}$. The regularization term in Eq. (2) will enforce the positive samples to be roughly the same distance away as all other negative pairings in the training triplets, while still minimizing their distance to the anchor tensors. However, if not careful, this approach could result in the model learning to map $n_i$ such that $d(\phi(a_i), \phi(n_i)) > \max\{\epsilon_0, d(\phi(p_i), \phi(n_i))\}$, which would ignore the triplet term, resulting in a minimization problem with no lower bound. To avoid such issues, we restrict $p = 2$ (or more generally, $p \equiv 0 \ (\text{mod } 2)$ ) as shown in Eq. (3):

$$\mathcal{L}_{Proposed}(p_i, a_i, n_i) = [d(\phi(a_i), \phi(p_i)) - d(\phi(a_i), \phi(n_i)) + \epsilon_0]^+ + [d(\phi(p_i), \phi(n_i)) - d(\phi(a_i), \phi(n_i))]^2. \quad (3)$$

Analytically, we can express the effect of regularization term and the rational behind allowed values of $p$ as the following:

$$\delta_+ \triangleq d(\phi_a, \phi_p); \quad \delta_- \triangleq d(\phi_a, \phi_n); \quad \rho \triangleq d(\phi_p, \phi_n).$$

With this notation, we can rewrite our proposed objective as:

$$\mathcal{L}_{Proposed} = \frac{1}{N} \sum_{(p_i, a_i, n_i) \in T}^{N} [\delta_+ - \delta_- + \epsilon_0]^+ + (\rho - \delta_-)^p.$$

$$(4)$$

Note that the since $[\delta_+ - \delta_- + \epsilon_0]^+ \geq 0, \mathcal{L}_{Proposed} = 0$ *if and only if* the summation of each term is identically zero. This yields the following relation:

$$-(\rho - \delta_-)^p = [\delta_+ - \delta_- + \epsilon_0]^+ \quad (5)$$

which, when $p \equiv 0 \ (\text{mod } 2)$, is only valid if $\rho = \delta_-$, and given that $\delta_- > \delta_+ + \epsilon_0$, we arrive at $\rho > \delta_+ + \epsilon_0$ (considering the real solutions). As a result, the regularization term enforces that the distance between the positive and the negative to be at least $\delta_+ + \epsilon_0$, leading to denser clusters that are better separated from other classes in space.

**Network Architecture and Training Procedure**. Our transformtion $\phi$ is a neural network consisting of three fully connected layers, each followed by a nonlinear activation

(Parametric Rectified Linear Units [PReLU]) and probabilistic dropout layers. Let $x_i \in \mathbb{R}^{b \times n}$ denote the input ($b$ samples with $n$ features), $\mathbf{L}(m, q)$ denote a linear operator in $\mathbb{R}^{m \times q}$, $PReLU(\cdot)$ denote Parametric ReLU (a pointwise function), and $\mathbf{D}(q)$ denote a dropout layer with probability $q$, then our architecture can be written in pseudocode form as:

$$x_i \rightarrow PReLU(\mathbf{L}(n \times 512)) \rightarrow \mathbf{D}(0.1)$$
$$\rightarrow PReLU(\mathbf{L}(512 \times 256)) \rightarrow \mathbf{D}(0.1)$$
$$\rightarrow PReLU(\mathbf{L}(256 \times d)) \rightarrow \phi(x_i)$$

where $d$ represents the output dimension, in our case $d = 32$. During training, we tuned the hyperparameters via grid search on the validation set. Our search resulted in $\epsilon_0 = 1$ and initial $lr = 0.001$ with the Adam optimizer, and the architecture listed above. To improve learning, we employed an exponential learning rate decay ($\Upsilon = 0.95$) to decrease after every 50 epochs, starting after 500 epochs until epoch 800 (last epoch).

### B. Personalized Blood Biomarker Models

To predict future biomarker values from a single lab visit, we leverage our similarity-based embeddings in addition to current value of the marker of interest. The rational for including these embeddings is to inform the donwstream prediction model of the complex relationship between various blood markers and lifestyle factors learned by our deep metric learning approach. We define three downstream prediction (regression) models where each differs only in its input, as shown in Fig. 1(a), with *age* and *sex* used as common inputs to all models. For a biomarker of interest, let us denote the current value as $b_t$ and the future value as $b_{t+1}$. Using these notations, we define our prediction models as the following:

- **Marker of interest + all other biomarkers + lifestyle**: Using dempgraphics and all current biomarkers as well as lifestyle factors to predict $b_{t+1}$.
- **Marker of interest + deep-learned embeddings**: Leveraging DL embeddings, age, sex and $b_t$ to predict $b_{t+1}$.
- **Marker of interest + deep-learned embeddings + lifestyle**: Adding lifestyle factors to the previous model to predict $b_{t+1}$.

In order to showcase the improvements made by model inputs, particularly lifestyle and deep-learned embeddings, and to allow for valid benchmarking of our results with the current state-of-the-art approach, we use the same XGBoost algorithm presented in Cohen *et al.* [2], including all same parameters, learning objective, and five-fold cross-validation strategy.

### C. Data and Data Pre-Processing

To show the potential of our approach on a broad population, we apply our methodology to the United Kingdom Biobank (UKB), a large-scale dataset consisting of EHR. UKB contains deep genetic and phenotypic data from approximately 500,000 individuals between 39-73 years, collected over many years across Great Britain. Initial assessments were collected between 2006 and 2010 which included a self-administered questionnaire, blood biomarkers (through blood biochemistry), and anthropometric measurements (Fig. 1(b)). UKB data also includes subsequent follow-ups where a subset of patients is invited for a repeat blood test assessment. The follow-up blood test measurements were done between 2012-2013 on approximately 20,000 participants from the first assessment (with many measurements missing, see Fig. 1(b)). Given our goal of predicting future biomarkers from a single visit, we utilize the follow-up assessment as our *prospective validation* data. That is, we only included participants' first visits for training regression models, and used the 2012-2013 follow-ups to evaluate the biomarkers predictions.

Given the complexity of the UKB and the scope of this research, we subset data to include participants' age and sex (demographics), all available blood biomarkers (that passed our quality assessment, described below), Metabolic Equivalent Task (MET) scores for physical activity, and self-reported hours of sleep and physical activity (complete list of features are provided in Fig. 1(c)). We leveraged both ICD-10 code and self-reported diagnosed conditions as well as current medication to assess participants' health, subsequently assigning each participant a label. After selecting these features, we ensured all features are at least 75% complete, dropping any features that did not meet this conditions. We then removed all patients with any null values. Then, we split the resulting data according to biological sex (male or female), and performed quantile normalization.

For each sex, we labeled patients based on their diagnosed condition or medication (e.g. participants with diabetes) or as the "apparently healthy" population (who do not have any serious health conditions and do not take illness-related medications). Due to age distribution imbalance, and to keep participants with similar ages close, we found the best distribution of individuals when age ranges were grouped unevenly. Each age group was constructed to approximate uniformity in the number of individuals while considering biological differences, with the age groups being *[36, 45], [46, 50], [51, 55], [56, 60], [61, 65], [66, 75]*. Additionally for the *prospective validation cohort*, we removed individuals whose repeat assessment were shorter than 2 years or longer than 5 years from the first visit. We then found the overlapping set of individuals who appear in both visits (after processing), ensuring that the first visit of these individuals *do not* appear in the training set of our representation learning model.

To stratify individuals based on lifestyle for our statistical analyses, following the American Heart Association guidelines, we considered 150 minutes of moderate activity or 75 minutes of vigorous activity per week as "active" lifestyle. Using the Metabolic Equivalents (METS) fields for moderate and vigorous activity, we placed a simple binary filter to classify individuals as "active" if they met the sufficient activity thresholds, or "less-active" otherwise. For sleep, we took the median hours of sleep in each age group as the reference value, and divided participants into those who sleep greater than or equal to the median hours (called "median sleep" group) or those who sleep less than the median hours of the group (labeled as "less sleep" individuals).

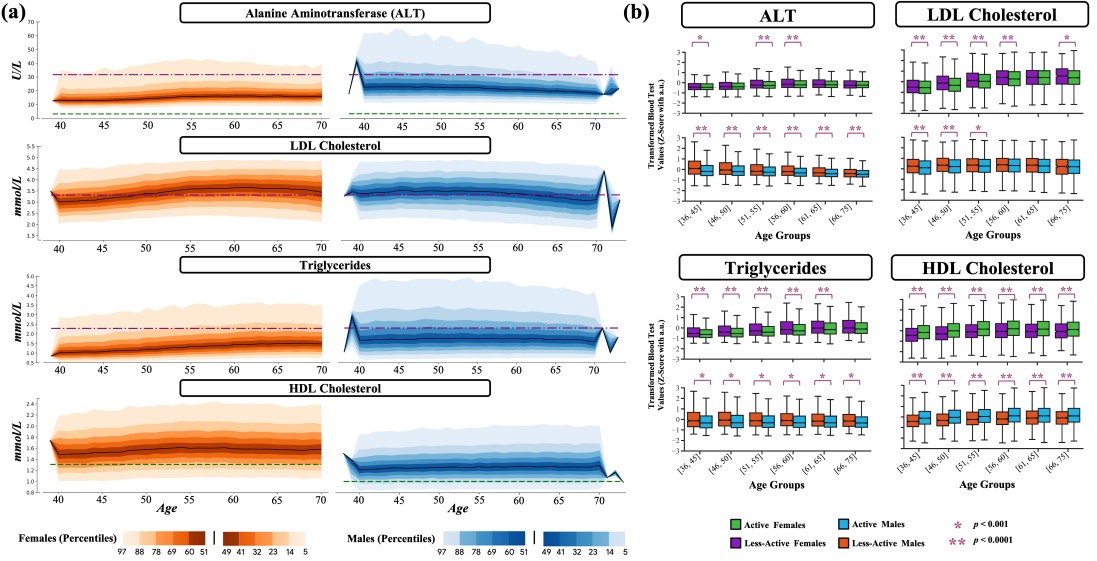

Fig. 2. **Differences in biomarker values based on sex, age, activity levels.** (a) Distribution (percentiles) of selected lab trends per sex. The $x$-axis represents age for for females (left, shades of orange) and males (right, shades of blue), with the median value highlighted as a black line. Clinical recommended ranges are marked for reference (upper and lower bound represented by dashed purple and green lines, respectively). (b) Result of performing statistical analysis between active and less active individuals (among the currently-healthy group) on a subset of biomarkers. Our results show that many blood biomarker distributions are statistically significantly different based on activity levels. Abbreviations: a.u. stands for arbitrary units after population-wide z-score normalization.

**Creation of Triplets and Inference Data.** To identify conditions (labels), we considered diagnosis and related medication. Due to the large number of diagnoses, we selected a subset of all conditions, prioritizing cardiometabolic conditions and co-morbidities based on their prevalence: Our selected conditions were *"Diabetes", "Diabetes and Cardiovascular", "Diabetes and Other Serious Conditions", "Diabetes, Cardiovascular and Other Serious Conditions", "Multiple Conditions (non-metabolic)", "Cardiovascular (not having Diabetes)", "Cardiovascular and Other Serious Conditions (not Diabetes)", "Respiratory", "Cancer".* All individuals with no diagnosis or medications were first labeled as *"Apparently Healthy"*. Using traditional reference ranges, we labeled an apparently-healthy individual as *bona fide* healthy if all blood biomarkers were within the traditional recommended ranges. The above steps resulted in 12 disease groups.

For modeling, we separated individuals based on biological sex and randomly selected 70% of each sex for training and 10% of participants for validation, ensuring that individuals with repeat assessment do not appear in these two sets. We used the remaining 20% of data for testing. From the training data for each sex, we created 100,000 unique triplet pairs randomly selecting positive and anchor points from the same conditions, and negative pairs from a different condition. We repeated these steps five times for evaluating our results.

## III. RESULTS

### A. Association of Biomarker Values and Lifestyle

To quantitatively test the importance of physical activity and sleep on blood biomarker ranges, we compared the distributions of participants stratified by sex (male or female), age, and lifestyle. Given the general Gaussian assumption on blood biomarker distributions and our empirical analysis, we performed Student's $t$-test to measure the difference in each age-sex group based on activity levels for 30 markers (highlighted with asterisks in Fig. 1(c)). In order to reduce type

I error, we adjusted our $p$-value using Benjamini-Hochberg correction, with a false discovery rate of $q = 0.05$. For female participants, our analysis showed that 22 out of the 30 ($\approx$73%) blood metrics were significantly different in at least one of the age groups, with 18 of 30 metrics ($\approx$60%) being significantly different in at least half of all age groups. Similarly, for male patients, we found that 19 of 30 metrics ($\approx$63%) were significantly different in one age group or more, with 18 being different in at least half of the age groups ($\approx$60%). We present our results for a subset of common blood biomarkers in Fig. 2(b). Similarly, we performed a $t$-test (with the same $p$-value adjustment as before) for detecting significant differences in the age-sex groups based on sleep levels for the same 30 biomarkers. Our analysis found 10 of 30 markers ($\approx$33%) in females and 11 of 30 ($\approx$36%) in males to be significantly different in at least one age group, 3 of 30 blood biomarkers (10%) and 4 of 30 ($\approx$13%) to be significantly different in at least half of age groups for females and male. Our analyses indicate that considering lifestyle, especially physical activity, in addition to age and sex, may improve personalized predictions and downstream tasks.

### B. Deep Representation Learning Improves Downstream Tasks

Using each individual's demographics, blood tests, and lifestyle signals as inputs (a total of 64 features), we trained our model to minimize the distance between individuals with the same conditions with respect to an anchor point, leveraging each person's existing conditions as labels to separate them from one another in space. As a qualitative assessment, we compared raw subject representations with our proposed embeddings in lower dimensions (using Uniform Manifold Approximation and Projection [UMAP]), which showed stark differences in individual's spatial distribution based on their conditions (Fig. 3(a)). Moreover, even though we did not explicitly express the relationship between comorbidities, our model was able to learn the similarities between these people

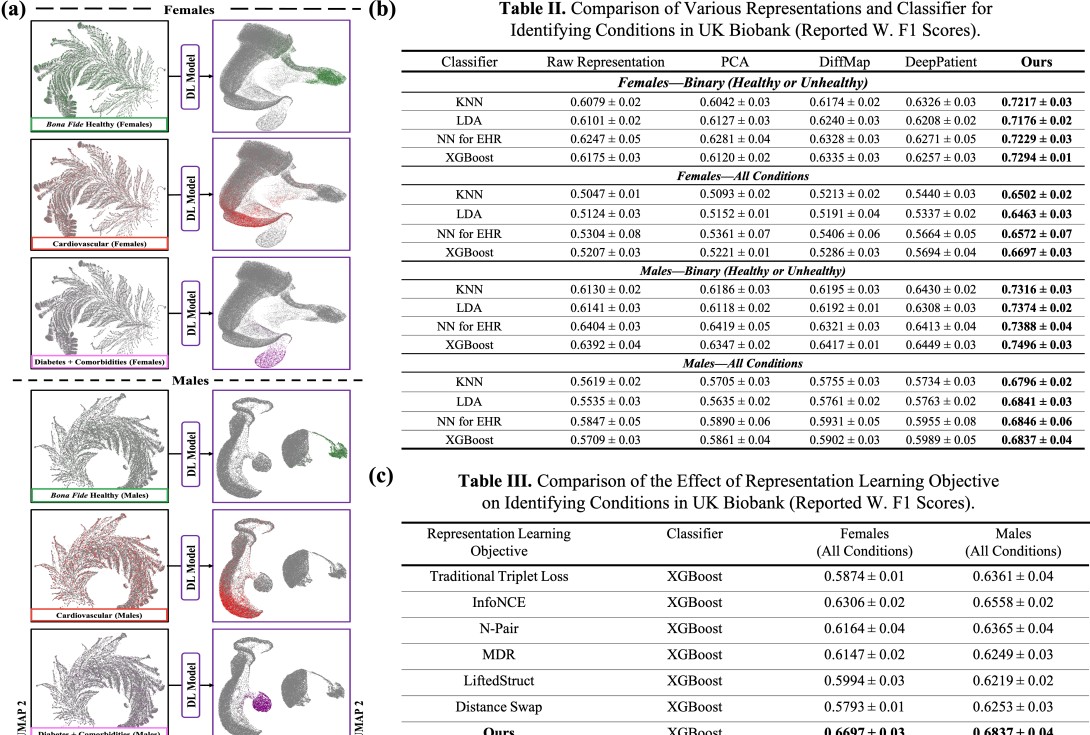

**Table II.** Comparison of Various Representations and Classifier for Identifying Conditions in UK Biobank (Reported W. F1 Scores).

| Classifier | Raw Representation | PCA | DiffMap | DeepPatient | **Ours** |
|---|---|---|---|---|---|
| *Females—Binary (Healthy or Unhealthy)* | | | | | |
| KNN | 0.6079 ± 0.02 | 0.6042 ± 0.03 | 0.6174 ± 0.02 | 0.6326 ± 0.03 | **0.7217 ± 0.03** |
| LDA | 0.6101 ± 0.02 | 0.6127 ± 0.03 | 0.6240 ± 0.03 | 0.6208 ± 0.02 | **0.7176 ± 0.02** |
| NN for EHR | 0.6247 ± 0.05 | 0.6281 ± 0.04 | 0.6328 ± 0.03 | 0.6271 ± 0.05 | **0.7229 ± 0.03** |
| XGBoost | 0.6175 ± 0.03 | 0.6120 ± 0.02 | 0.6335 ± 0.03 | 0.6257 ± 0.03 | **0.7294 ± 0.01** |
| *Females—All Conditions* | | | | | |
| KNN | 0.5047 ± 0.01 | 0.5093 ± 0.02 | 0.5213 ± 0.02 | 0.5440 ± 0.03 | **0.6502 ± 0.02** |
| LDA | 0.5124 ± 0.03 | 0.5152 ± 0.01 | 0.5191 ± 0.04 | 0.5337 ± 0.02 | **0.6463 ± 0.03** |
| NN for EHR | 0.5304 ± 0.08 | 0.5361 ± 0.07 | 0.5406 ± 0.06 | 0.5664 ± 0.05 | **0.6572 ± 0.07** |
| XGBoost | 0.5207 ± 0.03 | 0.5221 ± 0.01 | 0.5286 ± 0.03 | 0.5694 ± 0.04 | **0.6697 ± 0.03** |
| *Males—Binary (Healthy or Unhealthy)* | | | | | |
| KNN | 0.6130 ± 0.02 | 0.6186 ± 0.03 | 0.6195 ± 0.03 | 0.6430 ± 0.02 | **0.7316 ± 0.03** |
| LDA | 0.6141 ± 0.03 | 0.6118 ± 0.02 | 0.6192 ± 0.01 | 0.6308 ± 0.03 | **0.7374 ± 0.02** |
| NN for EHR | 0.6404 ± 0.03 | 0.6419 ± 0.05 | 0.6321 ± 0.03 | 0.6413 ± 0.04 | **0.7388 ± 0.04** |
| XGBoost | 0.6392 ± 0.04 | 0.6347 ± 0.02 | 0.6417 ± 0.01 | 0.6449 ± 0.03 | **0.7496 ± 0.03** |
| *Males—All Conditions* | | | | | |
| KNN | 0.5619 ± 0.02 | 0.5705 ± 0.03 | 0.5755 ± 0.03 | 0.5734 ± 0.03 | **0.6796 ± 0.02** |
| LDA | 0.5535 ± 0.03 | 0.5635 ± 0.02 | 0.5761 ± 0.02 | 0.5763 ± 0.02 | **0.6841 ± 0.03** |
| NN for EHR | 0.5847 ± 0.05 | 0.5890 ± 0.06 | 0.5931 ± 0.05 | 0.5955 ± 0.08 | **0.6846 ± 0.06** |
| XGBoost | 0.5709 ± 0.03 | 0.5861 ± 0.04 | 0.5902 ± 0.03 | 0.5989 ± 0.05 | **0.6837 ± 0.04** |

**Table III.** Comparison of the Effect of Representation Learning Objective on Identifying Conditions in UK Biobank (Reported W. F1 Scores).

| Representation Learning Objective | Classifier | Females (All Conditions) | Males (All Conditions) |
|---|---|---|---|
| Traditional Triplet Loss | XGBoost | 0.5874 ± 0.01 | 0.6361 ± 0.04 |
| InfoNCE | XGBoost | 0.6306 ± 0.02 | 0.6558 ± 0.02 |
| N-Pair | XGBoost | 0.6164 ± 0.04 | 0.6365 ± 0.04 |
| MDR | XGBoost | 0.6147 ± 0.02 | 0.6249 ± 0.03 |
| LiftedStruct | XGBoost | 0.5994 ± 0.03 | 0.6219 ± 0.02 |
| Distance Swap | XGBoost | 0.5793 ± 0.01 | 0.6253 ± 0.03 |
| **Ours** | XGBoost | **0.6697 ± 0.03** | **0.6837 ± 0.04** |

Fig. 3. **Qualitative and quantitative results of our proposed metric learning on UKB. (a)** UMAP visualization of the untransformed space (left column) compared to the UMAP visualization of the learned embedding space through our proposed model (right column), both for female (top) and male (bottom) participants in the UKB. **(b)** Comparison of commonly-used representations for EHR, namely PCA, Diffusion Maps (DiffMap), DeepPatient, and our proposed representation learning. To show the effect of the representations as opposed to the classification schemes, we use four different classifiers (K-Nearest Neighbors [KNNs], Linear Discriminant Analysis [LDA], Neural Network (NN) for EHR [27], and Extreme Gradient Boosting Ensemble [XGBoost]). Boldface values indicate the highest accuracy in terms of weighted F1 scores. **(c)** Comparison of our proposed metric learning objective with commonly-used metric and contrastive learning objectives, namely InfoNCE [28], N-Pairs [29], Multi-Level Distance Regularization (MDR) [30], LiftedStruct [31], and Distance-Swap Triplet Loss [32].

and embed them closer to one another (e.g., see *Diabetes + Comorbidities* panels in Fig. 3(a)).

To quantitatively assess the quality of various embeddings, we started by performing a binary classification of whether any health conditions exist, *i.e.* separating people based on those with doctor-confirmed conditions (or medications) and those without any conditions or condition-specific medications. For traditional approaches, we identified Principal Component Analysis (PCA) (linear transformation), and Diffusion Maps (DiffMap) (nonlinear transformation) as two of the most common transformations used for representing EHR data. Using these methods, we transformed the UKB data and classified individuals as "healthy" or "unhealthy" using four common classifiers to show the true effect of embedding on identifying conditions. Our results, presented in Fig. 3(b)-Table II, indicate that our model's embeddings significantly improve classification accuracy across all tested classifiers (over 10% improvement in weighted F1 score for both males and females). To further validate our results, we also trained each of the classifiers to predict the underlying conditions (twelve aggregated conditions in total), thus constituting a multi-class classification. Similar to the binary case, our deep-learned embeddings enabled significant improvements in classifying different conditions, with over 9% improvement compared to the next-best approach (Figure 3(b), Table II). These results indicate the tremendous potential of transforming raw blood biomarkers with our proposed apporach, which can facilitate and enhance various downstream tasks. To further demonstrate the effectiveness of our novel metric learning approach, we compared our model with current state-of-the-art metric learning models on the UK Biobank for the same classification tasks. Our results showed our approach outperforming networks trained with common metric learning objectives and contrastive architectures (loss function and data curation), as shown in Fig. 3(c)-Table III, thus showing the improvements of our proposed objective.

*C. Personalized Blood Biomarker Model for Future Value Prediction*

To measure the accuracy of our future lab results prediction, we calculated the $R^2$ values of our prediction on 6440 apparently-healthy individuals in the prospective validation cohort (using the first visit to predict the next blood biomarker values), with the resulting $R^2$ ranging from 0.05 to 0.81. For 43 of 55 biomarkers, we found that adding lifestyle factors alone improved $R^2$ scores over the baseline (marker of interest only) and the current best model for single time prediction

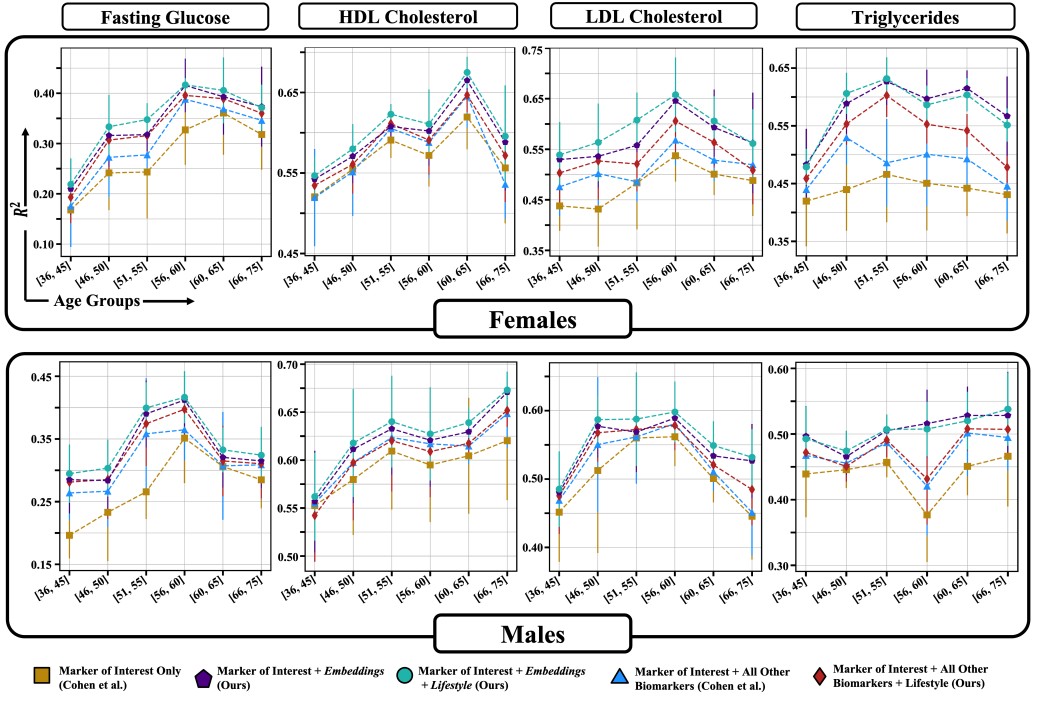

Fig. 4. **Future biomarker prediction results on the UK Biobank prospective validation cohort**. Here we present the average predictions accuracy ($R^2$) and standard deviation across five-fold cross validation for our proposed models, as well as the current single-time state-of-the-art models introduced by Cohen *et al.* on four randomly-selected metabolic biomarkers. Training and testing of future biomarker prediction models was done using UK Biobank participants who had repeat assessments within two to five years after first UK Biobank blood test (who were excluded from the training set of our representation learning step). Top panel depicts the results for female participants and bottom panel shows the result for male participants

(marker of interest + all other biomarkers by Cohen et al's) by over 5% in *at least* half of the age groups, with notable enhancements among the metabolic blood biomarkers. Additionally, our results indicated that the addition of our proposed deep-learned embeddings further increased $R^2$ scores in 50 of 55 blood biomarkers compared to the other strategies by over 10%. Moreover, we found that adding lifestyle factors directly to the "Marker of interest + Embeddings" model (resulting in the "Marker of interest + Embeddings + Lifestyle" model) achieved the best performance in at least half of the age groups for 47 of 55 tested laboratory markers. We present our results for four randomly-selected metabolic markers in Fig. 4.

## IV. CONCLUSIONS AND DISCUSSION

In this work, we introduced a framework for personalized blood biomarker models which aim to move beyond population-level statistics, and incorporating individual lifestyle and demographics factors that significantly impact and personalize these reference values. We demonstrated that lifestyle, particularly physical activity levels, plays a crucial role in shaping blood biomarker distributions, often more prominently than even age or sex in specific cohorts. This highlights the limitations of population-level statistics and underscores the need for personalized approaches that account for individual variability.

We introduced a novel deep metric learning approach that aims to capture the complex interactions between biomarkers by learning a similarity-based representation of health data. Our method outperforms commonly used traditional representation learning techniques, *e.g.* PCA and Diffusion Map, and models trained with state-of-the-art contrastive and metric learning objectives, demonstrating our model's ability to capture clinically-relevant information in the produced embeddings. Furthermore, we showed that integrating these similarity-based embeddings into future biomarker prediction models lead to significant improvements in prediction accuracy, particularly for metabolic markers. The addition of lifestyle features directly into these models amplifies their predictive power, highlighting the critical role of lifestyle in shaping future biomarker values. Our results showcase the utility of our approach in a clinical setting where future value of blood biomarkers can be estimated using only a single laboratory visit. Additionally, our personalized approach is able to provide important context and interpretability for out-of-norm blood biomarker values using patients' history, which can be of significant value for preventive care and diagnosis in clinical applications.

A limitation of our work is the lack of large longitudinal cohort that has continuous phenotypes and lifestyle dataset, where we can train models to predict future outcome at specific time horizon. We used the UK Biobank, which currently is the largest available dataset (500K participants), where only 4% of participants (20K) have a second follow-up blood test. Hence, our approach was designed specifically for single-time representation learning and prediction. However, the inclusion of additional timepoints and visits, when available, can potentially provide valuable context and insights, a feature that is not currently possible with our framework. Additionally, UK Biobank is less diverse cohort with majority from European ancestry. We hypothesize that our results can be further strengthened through training and validation on more diverse set of patient populations. In the future, we plan to include "All of Us" data [33], which consists of a more diverse cohort, in our training pipeline.

With the rise of continuous activity and sleep monitoring

through wearables (e.g., Fitbit, Apple Watch, Google Pixel, etc.), there is a tremendous opportunity to utilize individuals' lifestyle information for improving preventive interventions, diagnosis and care. Integration of EHR data with wearable-derived lifestyle metrics at scale can allow for implementation of more personalized lifestyle-informed health models, such as the one introduced in this study. Moreover, the additional digital biomarkers and the granularity of wearable-derived lifestyle factors have the potential to further enhance the prediction of personalized biomarker references. By enabling the prediction of personalized blood biomarker values from a single patient visit, our framework has the potential to empower (1) *early disease detection and risk stratification*, since personalized reference ranges provide a more accurate baseline for individual patients, enabling earlier detection of deviations that may indicate underlying health risks, (2) *Tailored interventions and preventative care* by understanding the influence of lifestyle on individual biomarker trajectories, and (3) *improved patient outcomes* through leveraging advanced ML models to better analyze EHR, holding the potential to improve diagnostic accuracy and optimize treatment strategies. Our findings pave the way for a new paradigm in healthcare where individual variability and lifestyle are no longer seen as confounding factors, but as essential components for personalized care and treatments.

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
