# OpenReview forum: "Lifestyle-Informed Personalized Blood Biomarker Prediction via Novel Representation Learning"
_IEEE.org/EMBS/BHI/2024/Conference — IEEE BHI'24_

### Official Review · Reviewer_yRRM · 2024-08-11
**Paper motivated and written well,**

**Overall Rating:** 7
**Confidence:** 4

**Other Quality Metrics:**

Clarity of writing: great
Clinical Significance: Excellent
Methodological Novelty: good
Experiments and Results: Excellent

**Questions For The Authors:**

Please refer to the Weaknesses.

**Strengths:**

* Paper is very well-written and well-thought-out.
* The results are comprehensive.

**Summary Of The Paper:**

Blood biomarkers such as ALT and Cholestrol have profound effects on diagnosing and monitoring medical conditions. This paper moves in the direction of making more accurate estimations of such biomarkers through the use of lifestyle data and previous blood biomarkers. They achieve it through two steps: learning a representation of individuals, and then using that representation along with the previous biomarker reading to predict the next one.

**Weaknesses:**

* I was a bit lost in the motivation, the start of the intro. The paper introduces a motivation of the paper to be having more accurate recommended ranges. But why should the recommended range be so much personalized? If by recommended range, we mean the range that the patient's blood biomarker needs to be in, why shouldn't it be instead driven by the population, to signify what the healthy value is? And this more specifically about issues such as lifestyle, and less about characteristics such as age.
* The part on learning patient similarity, although interesting and sound, could be connected to the main story better.

---

> ### Author Rebuttal · Authors · 2024-08-31
>
> We thank the reviewer for their insightful review and suggestions.
>
> We agree with the reviewer that reference/recommended ranges should be driven by the broader healthy population. In addition to using these normal (healthy) ranges, our approach aims to provide a personalized reference that can be leveraged to better assess patients' health trajectories and prevent future complications, even if the biomarker values are currently within the population-level normal values.
>
> As an example for the value of personalized references, we can consider a real UKB participant (EID 4371531) who, at the time of first visit, had reported no serious cardiovascular conditions with normal total cholesterol (4.6 mmol/L), normal triglycerides (1.58 mmol/L), normal HbA1c (32.9 mmol/mol), and elevated LDL at 2.9 mmol/L. Our model, using the learned representations of raw features (which include age), lifestyle, and the current biomarker value, predicts triglyceride and total cholesterol to be outside of the *personalized reference* at the time of next visit (predicted values of 2.2 mmol/L and 5.7 mmol/L, respectively), though these values were within the population-level normal ranges at the time of first visit. In the next follow-up visit years later, this individual reported to have had a stroke, and their blood test show elevated triglycerides value of 2.09 mmol/L. We believe that having personalized blood test references will be especially valuable for such cases, where by using personalized models healthcare providers can suggest preventative measures at the time of the first visit to reduce the risk of future health complications. We will add the example above and relevant explanations to the camera-ready version of the manuscript, if accepted.
>
> We appreciate the reviewer's suggestion on improving the connection between the motivation of metric learning (“similarity”) and the main story. We have revised our manuscript to better emphasize on how learning patient similarities from the population can provide additional interpretations and personal insights regarding an individual’s blood test values that otherwise may not be available (e.g. personalized blood test references). These revisions are marked in fuchsia in the PDF.
>
> We thank the reviewer for their feedback. We believe that the suggested changes to the manuscript have improved the previous version of our paper. We would be happy to discuss additional questions.

---

### Official Review · Reviewer_egC6 · 2024-08-11
**Lifestyle-Informed Personalized Blood Biomarker Prediction via Novel Representation Learning**

**Overall Rating:** 7
**Confidence:** 1

**Other Quality Metrics:**

Clarity of Writing : Good
Clinical Significance : Good if the author empjasize setting  thresholds or ranges for biomarkers  taking into account age, sex and other  lifestyle facotrs
Methological Novelty : Novel Enough
Experimental Results : Satisfactory

**Questions For The Authors:**

None

**Strengths:**

Well written paper emphasizing life style factor's importance on setting limits or thresholds on biomarkers.  Indeed most biomarkers do not take into consideration age, sex and other physical activity.  The prediction of  future labs based on a single lab is a bit  challenge considering many factors effecting biomarkers.

**Summary Of The Paper:**

The paper  explores the relationship between  biomarkers and life style factors and proposes deep learning embeddings  for predicting clinical diagnoses.  They propose a methods to predict future lab values from a single lab test and life style factors.

**Weaknesses:**

Prediction of future lab values based on a single lab  result  even with  the added l;ife style factors is a stretch considering the many factors that effect biomarkers and the variability of the biomarkers over time.

The generalizability of the work is limited due to the low rate of return visits in the database.  The data is biased  with only 4% of the population having a second lab result.

---

> ### Author Rebuttal · Authors · 2024-08-31
>
> We appreciate the reviewer’s valuable feedback on our manuscript.
>
> We agree with the reviewer that using multiple timepoints to predict future lab values may be more accurate; our proposed framework can be easily extended to include multiple timepoints for predicting personalized blood test references. However, the high patient churn rate and the importance of early interventions signify the need for models that can predict future outcomes with as little available information as possible, in our case in the first visit. We agree with the reviewer's point regarding many contributing factors (such as diet changes, developing health conditions, etc.) that can change blood test results from the time of first visit to the next. However, having the predicted personalized reference value from the first visit can provide the healthcare provider with valuable information and additional context around the individual’s overall health:
>
> For example, let us consider an individual with a current insulin value of 13 mcU/mL, which is normal (since <17 mcU/mL). Let us assume that at the time of the visit, using our approach, the healthcare providers can compute the predicted value of insulin to be at 19.5 mcU/mL. Having access to this information enables a clinician to provide the individual with necessary guidance on nutrition and lifestyle changes at the time of first visit, which can potentially avoid abnormal blood insulin levels that can lead to developing serious health conditions such as insulin resistance and diabetes.
>
> We thank the reviewer for raising the point on generalizability of our approach. As mentioned in the discussion section of the manuscript, we believe that our approach can be further improved by training/evaluating our model on other diverse datasets. We acknowledge that the relatively low percentage of participants of the UKB with follow-up lab tests presents a limitation in terms of generalizability to populations with different healthcare access patterns. We note that our testing set for the second visit still comprised over 6,000 participants, and while not fully representative of the entire population, this subset provides a significant sample size to develop and validate our initial model, which focuses on early intervention based on first-visit data.
>
> We thank the reviewer again for their questions and comments, and we would be happy to respond to further questions or make additional changes to our manuscript to improve the current draft.

---

### Official Review · Reviewer_eapm · 2024-08-13
**Lifestyle-Informed Personalized Blood Biomarker Prediction via Novel Representation Learning**

**Overall Rating:** 7
**Confidence:** 5

**Other Quality Metrics:**

Clarity of writing: Good
Clinical Significance: Great
Methodological Novelty: Excellent
Experiments and Results: Great

**Questions For The Authors:**

In terms of practical implementation, how do you see the proposed model being integrated into existing clinical workflows? What challenges do you foresee in collecting the necessary lifestyle data in real-world settings, and how might these challenges impact the model’s effectiveness?

**Strengths:**

The introduction of a novel deep metric learning framework is a key innovation. This framework effectively captures the complex relationships between biomarkers and lifestyle factors by creating a similarity-based embedding space. This method demonstrates superiority over traditional and existing state-of-the-art representation learning techniques, particularly in predicting clinical diagnoses.
One of the most promising aspects is the integration of lifestyle factors, such as physical activity and sleep, with blood biomarker data. This approach acknowledges the significant role that lifestyle plays in influencing health outcomes, moving beyond traditional population-level statistics that often neglect individual variability.
The ability to predict future blood biomarker values from a single lab visit, using the learned embeddings and current biomarker data, is a significant advancement. This capability has the potential to improve risk stratification and provide more tailored healthcare interventions, enabling earlier disease detection and more precise preventative care strategies.

**Summary Of The Paper:**

The paper discusses a deep learning framework that combines lifestyle data, like physical activity and sleep, with blood biomarkers to predict personalized future biomarker values. The goal is to overcome the limitations of traditional blood biomarker reference values, which often do not account for individual differences influenced by lifestyle and genetics. The authors propose a method for predicting future biomarker values and establishing personalized reference ranges using lifestyle data.

The key methodology involves a deep metric learning framework that creates a similarity-based embedding space, capturing the complex relationships between blood biomarkers and lifestyle factors. The authors validate their model using data from the UK Biobank, which includes 257,000 participants. Their approach aims to predict future blood biomarker values from a single lab visit by utilizing the learned embeddings alongside current biomarker values.

The study shows that deep-learned embeddings outperform traditional and state-of-the-art representation learning techniques in predicting clinical diagnoses. In a subset of 6,440 UK Biobank participants with follow-up visits, the model incorporating lifestyle factors and learned embeddings significantly improves the prediction of future lab values compared to models that do not consider lifestyle data. The results demonstrate that including lifestyle factors in blood biomarker models leads to more accurate predictions, underscoring the importance of lifestyle considerations in clinical settings.

The authors conclude that their framework provides a basis for more accurate risk stratification tools and personalized healthcare strategies. They emphasize the critical role of personalized blood biomarker references in early disease detection and tailored preventive care. However, they acknowledge certain limitations, such as the lack of large longitudinal datasets with continuous lifestyle data and the demographic homogeneity of the UK Biobank cohort. Despite these limitations, the study represents a significant advancement in personalized healthcare by integrating lifestyle data with biomarker analysis to improve the prediction of future health outcomes. This approach has the potential to enhance diagnostic accuracy and optimize treatment strategies by moving beyond population-level statistics to more individualized healthcare solutions.

**Weaknesses:**

The paper does not sufficiently address how the proposed model could be scaled and implemented in real-world clinical settings. There are practical considerations, such as how easily lifestyle data can be collected and integrated into electronic health records (EHRs), and whether the computational demands of the model are feasible in a clinical environment. The authors could enhance their work by discussing these challenges and potential solutions, such as integrating the model into existing EHR systems or developing lightweight versions of the model that can be used in resource-constrained settings.

---

> ### Author Rebuttal · Authors · 2024-08-31
>
> We thank the reviewer for their insightful review and raising the point regarding scalable implementation of our approach in a clinical setting.
>
> We believe that the prevalence of wearable health trackers and smartwatches (e.g. Fitbit , Apple, etc.) can provide accurate, continuous, and scalable tracking of lifestyle features that can be used in tandem with existing blood test data for our proposed method. Recent research shows that roughly 1 in 3 US adults wear a health tracker capable of computing lifestyle metrics (e.g. daily steps), with more than 80% indicating a willingness to share device information with their doctor to support their health monitoring [[Source 1](https://jamanetwork.com/journals/jamanetworkopen/fullarticle/2805753)]. Incorporating wearables data can provide a very scalable and cost-effective solution for using lifestyle-informed health models, such as our proposed approach, in clinical settings. We acknowledge the broad adoption in the marketplace of wearable devices has not yet translated into that data being used in clinical practice. However, efforts to integrate this data with clinical records (e.g. Apple Health App) will hopefully aid with wider adoption by healthcare providers. Additionally, multiple insurance companies are working on wider adoption of such integration, for example, United Health pulling wearables data to their systems and rewarding participants who move more than 12000 steps a day [[Source 2](https://www.unitedhealthgroup.com/newsroom/2018/2018-11-15-apple-watch-uhc-wearable.html)]. Once the blood test data and lifestyle factors are integrated, our proposed model can be deployed on the system (our approach is lightweight and can be run on-device) or in the cloud to provide healthcare providers with personalized blood test references.
>
> We agree with the reviewer that adding details on potential paths for implementing our proposed approach in a clinical setting would improve the quality of our work, and have added a summary of above discussion to the last paragraph of the discussion section marked with purple in the revised PDF (revised text not included here due to character limit).
>
> We would like to thank the reviewer for their comments and questions. We hope that our explanation above and the changes to the manuscript have adequately addressed the reviewer's questions. We would be happy to answer any additional questions or to make additional changes to further improve our manuscript.

---

### Decision · Program_Chairs · 2024-09-23

Accept